# Non-Traditional Pro-Inflammatory and Pro-Atherosclerotic Risk Factors Related to Systemic Lupus Erythematosus

**DOI:** 10.3390/ijms232012604

**Published:** 2022-10-20

**Authors:** Patricia Richter, Anca Cardoneanu, Ciprian Rezus, Alexandra Maria Burlui, Elena Rezus

**Affiliations:** 1Department of Rheumatology, University of Medicine and Pharmacy “Grigore T Popa”, 700115 Iasi, Romania; 2Clinical Rehabilitation Hospital, 700661 Iasi, Romania; 3Department of Internal Medicine, University of Medicine and Pharmacy “Grigore T Popa”, 700115 Iasi, Romania; 4“Sfantul Spiridon” Emergency Hospital, 700111 Iasi, Romania

**Keywords:** atherosclerosis, systemic lupus erythematosus, inflammation, cytokines

## Abstract

Cardiovascular diseases (CVD) are one of the leading causes of high mortality in patients with systemic lupus erythematosus (SLE). The Framingham risk score and other traditional risk factors do not fully reflect the CVD risk in SLE patients. Therefore, in order to stratify these high-risk patients, additional biomarkers for subclinical CVD are needed. The mechanisms of atherogenesis in SLE are still being investigated. During the past decades, many reports recognized that inflammation plays a crucial role in the development of atherosclerosis. The aim of this report is to present novel proinflammatory and pro-atherosclerotic risk factors that are closely related to SLE inflammation and which determine an increased risk for the occurrence of early cardiovascular events.

## 1. Introduction

Systemic lupus erythematosus (SLE) is an inflammatory autoimmune disease with the involvement of various organs [1]. It has been demonstrated that cardiovascular diseases (CVD) are currently a leading cause of comorbidity and mortality in SLE patients [2,3,4,5]. Particularly, coronary artery disease (CAD) causes nearly 30% of deaths in SLE [2,6,7]. SLE patients present a great prevalence of asymptomatic CAD [8]. Young women with SLE are prone to higher risks, as they are 50 times more likely than controls to develop a myocardial infarction; nevertheless, males also present the worst outcomes [9,10,11,12]. Regarding the risk of cerebrovascular complications in SLE, it is lower than coronary diseases, but compared to the general population, the risk is twice as high [2,13]. If these occur later in the disease course, the main cause may be attributed to atherosclerosis [14]. Additionally, the risk of peripheral artery occlusive disease has been found to be nine-fold higher in SLE patients than in general population [2,15].

The main traditional risk factors involved in CVD are: older age, arterial hypertension, obesity, and high cholesterol serum levels [2,16,17,18]. Particularly, for almost a century, cholesterol has been regarded as the main factor that promotes the development of atherosclerosis [19]. The burst of cardiovascular events (CVE) in SLE cannot be entirely related to traditional risk factors; it is thought to be driven also by immunologic and inflammatory features found in SLE [9,20,21,22]. Unfortunately, all these traditional cardiovascular models in use underestimate the risk of fatal events in SLE patients. There are currently no lupus-specific screening investigations to classify patients at risk for future major CVE, although traditional Framingham risk factors are insufficient to assess the risk of CVD [9,23,24]. Thus, extensive studies regarding non-traditional SLE-related risk factors that contribute to cardiovascular complications have been conducted [25].

Studies and book chapters relevant to this work were identified using the PubMed, EMBASE, Web of Science and Clinical Key research platforms. We have selected the most important works published in the last 5 years (2017–August 2022). For certain significant data for this review, but for which we found insufficient data in this time period, we extended the search to the last 10 years (2012–2022).

## 2. Atherosclerosis and Inflammation in SLE

Subclinical atherosclerosis, an early event in SLE individuals, plays an important role related to cardiovascular risk and morbidity. Among the changes it causes, we can list peripheral embolism, arterial hypertension with subsequent left ventricular hypertrophy, and diastolic dysfunction [2,26,27].

Medical conditions characterized by systemic inflammation like SLE have been strongly associated with atherosclerosis. Evidence show that inflammation process is crucial for the development of accelerated atherosclerosis in SLE. Therefore, it is plausible to assume that SLE inflammation may consecutively increase the CVD risk [2,9,28,29]. Inflammation has been linked to the development of atherosclerotic CVD in the general population as well [9,30,31]. Reducing inflammation is one of the essential targets in decreasing the CVD risk in SLE patients; it was highlighted that inflammation is responsible for initiation, formation and destabilization of atherosclerotic plaques [19,25,32,33].

### Atherosclerosis Mechanism in SLE

Under a variety of irritant factors, endothelial cells on the vascular wall become activated; this results in a cascade of inflammatory reactions that can lead to the formation of atherosclerotic plaque [34]. There are three stages illustrated in Figure 1, as follows:

I. Firstly, the inflammatory process leads to the activation of the endothelial cells that determines the expression of surface molecules such as vascular cell and intercellular adhesion molecule-1 (VCAM-1, ICAM-1), selectins, and integrins necessary to leukocyte adhesion, rolling and attachment [2,35,36].

II. Secondly, the adherent leukocytes move through the intima layer and reach the media. This monocyte transmigration into subendothelial layer is promoted by chemokines, such as monocyte chemotactic protein-1 (MCP-1) [2,34,35]. Interestingly, a report showed that a high carotid intima-media thickness (IMT) is associated with elevated circulating levels of MCP-1 in humans [2,31]. The expression of MCP-1 and interleukin-8 (IL-8) can be induced by l-homocysteine, another factor that stimulates leukocyte recruitment [2,37].

At this point, tumor necrosis factor-α (TNF-α), interleukin-1 (IL-1), and oxidized low-density lipoprotein (OxLDL) play two significant roles: first, they upregulate the adhesion molecules, and second, the adhesion of MCP-1 [2,38].

III. Low-density lipoproteins (LDL) are trapped in the subendothelial region, where they are exposed to reactive oxygen species (ROS) and are transformed into OxLDL. This is the beginning of the third and final stage. OxLDL, in turn, amplifies the inflammatory response by activating endothelial cells that will secrete adhesion molecules and chemokines that will stimulate the recruitment of monocytes. Additionally, OxLDL, found in the subendothelial layer, stimulates monocytes to differentiate into macrophages. The conversion of macrophages into foam cells and the multiplication of smooth muscle cells occur in this last stage, leading to atherosclerotic plaque proliferation [2,34,35,39,40].

On the other hand, the interaction between platelets and endothelial cells generates the production of cytokines such as IL-8 and stimulates ICAM-1, leading to endothelial dysfunction and a high risk of thrombosis. In SLE, these platelet–endothelium interactions are frequently seen and can accelerate the development of CVE [34,41].

## 3. Non-Traditional Pro-Atherosclerotic Biomarkers in sSLE

Multiple non-Framingham inflammatory biomarkers, such as plasma soluble TNF-like weak inducer of apoptosis (sTWEAK), dysfunctional pro-inflammatory high-density lipoprotein (HDL) (piHDL), leptin, and homocysteine, are independently linked to atherosclerosis in SLE [9,42,43,44,45,46]. Due to the increasing number of individual biomarkers attributed to CVD risk in SLE, a study from the University of California, Los Angeles, has developed a biomarker panel that could help to quantify more accurately the atherosclerosis risk and CVE risk in SLE. Thus, a model was created that is called PREDICTS (the Predictors of Risk for Elevated Flares, Damage Progression, and Increased Cardiovascular Disease in SLE) [45]. It integrates four inflammatory biomarkers (homocysteine, piHDL, TNF-like weak inducer of apoptosis TWEAK, and leptin) and two risk factors (age and diabetes). Among non-traditional risk factors, an increased piHDL function, high leptin levels and plasma soluble TWEAK significantly correlated with plaque formation. Interestingly, this complete panel showed a better predictive capacity for plaque formation in SLE compared with individual markers [45].

### 3.1. piHDL

Although HDL cholesterol is typically regarded as atheroprotective, in systemic inflammation it can change its properties. This occurs when it shifts from the typical anti-inflammatory form to the proinflammatory HDL (piHDL) form, phenomena that tend to occur during the onset of chronic inflammatory conditions, also seen in SLE [2,47]. A cause may be the ongoing oxidative damage in the chronic inflammatory state that generates piHDL and high levels of ox-LDL [48]. piHDL is found in SLE patients more frequently compared to rheumatoid arthritis (RA) individuals and the general population (Table 1). In this situation, the favorable effects of HDL, the decrease of LDL oxidation, are lost. Instead, this dysfunctional HDL enhances accelerated atherogenesis. McMahon et al. showed that piHDL is strongly linked to the development of carotid plaque and intima-media thickness (IMT) in SLE patients [34,45,48,49,50]. Interestingly, dysfunctional HDL may also be generated by atypical HDL oxidation. This is driven by the release of neutrophil extracellular traps (NETs) and low-density granulocytes that are specific for SLE [9,31,51]. Moreover, piHDL has proteomic and lipidomic abnormalities that are specific for HDL particles derived from SLE individuals [9,52,53,54].

Summarizing, it seems that piHDL is commonly found in SLE patients compared to general population. There is a strong link between piHDL and carotid plaque progression. In SLE cases, piHDL has atherogenic properties due to oxidative damage and also due to proteomic and lipidomic abnormalities.

### 3.2. Endothelial Progenitor Cells

Endothelial dysfunction, seen in the early stages of atherosclerosis development, has been studied for its contribution to CVD risk in SLE [25,56,57,58]. Data suggested that endothelial dysfunction is already present in SLE patients before CVD development [25,59,60,61,62]. Patients with SLE have functionally and quantitatively decreased levels of endothelial progenitor cells (EPCs) [25,63]. EPCs are a subpopulation of circulating stem cells involved in the repair of blood vessels [34]. They are essential for maintaining the endothelial function, neovascularization, and vascular repair [25]. EPCs are thought to be able to replace damaged endothelial cells and restore endothelial integrity and, therefore, to improve the endothelial function [25,29,64]. They might be used as possible biomarkers to identify patients at risk for developing a CVD [34,65]. Castejon et al. showed that SLE patients with lower levels of progenitor cells had an increased arterial stiffness and a greater preponderance of cardiovascular risk factors, such as smoking and metabolic syndrome [34,66].

In multiple reports evaluating patients with SLE versus healthy controls, the levels of circulating EPCs were significantly low (Table 2). Thus, a reduced level of circulating EPC may be a marker for an early vascular disease [34,66]. The therapeutic and prognostic role of EPCs as an early CVD biomarker in SLE patients remain to be elucidated. New methods for measuring endothelial function and quantifying EPC as prognostic indicators of CVD are needed in SLE patients [25].

Encouraging in vitro results show the reversibility of the decrease in EPC number after treatment with anti-type 1 INFα and anti B cell activating factor (BAFF) [34].

In summary, it is known that SLE patients present endothelial dysfunction more frequently. It seems that an important biomarker can be considered the low level of EPCs. Moreover, these cells have an altered function. The decrease in the number and alteration of the functionality of EPCs is associated with an increase in arterial stiffness and with the occurrence of CVD in SLE cases.

### 3.3. Endocan

Endocan is another marker of angiogenesis and endothelial cell activation. It contributes to the recruitment, adhesion and migration of leukocytes across the endothelium [2,74,75].

Icli et al. investigated the impact of endocan on atherosclerotic process in SLE. They found that endocan serum levels were higher in SLE patients versus controls; also, endocan levels were strongly associated with carotid intima-media thickness (cIMT) [2,76]. Its potential role as a useful biomarker in the SLE pathogenesis needs to be further investigated (Table 3). In the literature, only two studies focus on endocan in SLE and on the link between it and atherosclerosis. Furthermore, no animal model studies were identified.

In conclusion, although there are only few studies focused on endocan role in atherosclerosis, it has been found an increased endocan titer in SLE individuals versus controls. Moreover, it seems that there is an association between the endocan level and carotid intima-media thickness. These studies raise the hypothesis that endocan may become a reliable predictor of early atherosclerosis in SLE patients.

### 3.4. Leptin

Leptin, an adipokine that controls satiety and fat deposits, is considered to be an atherosclerotic risk factor [34,78]. According to the first study that examined the relationship between adipokines and subclinical atherosclerosis in SLE individuals, McMahon et al., showed that higher leptin levels were independently associated with carotid plaques formation and positively correlated with piHDL and oxidized phospholipids levels (Table 4) [34,44,79].

Many immune cell subsets are influenced by leptin [9,80]. For instance, leptin may have particularly pro-inflammatory actions on macrophages in SLE, such as the activation of phagocytosis and an increased presentation of apoptosis-derived self-antigen to T lymphocytes [9,81]. In addition, leptin may induce an increased production of pro-inflammatory cytokines and oxidative stress in endothelial cells and cardiomyocytes [9,82,83].

However, some contradictory studies reported a reduced leptin serum in SLE patients [84,85], while others reported no statistically significant differences between SLE patients and controls [86,87].

**Table 4 ijms-23-12604-t004:** Leptin and SLE patients.

Biomarker	Study	Results
**Leptin**	Garcia-Gonzalez et al., 2002 [88]	Elevated leptin levels in women with SLE than control
	Sada et al., 2006 [89]	Higher plasma leptin levels in SLE patients than controls
	Chung et al., 2009 [90]	No important correlation between leptin levels and coronary calcification in SLE individuals versus controls
	Al et al., 2009 [91]	Elevated serum leptin concentrations in one-third of SLE patients
	Kim et al., 2010 [92]	Increased leptin levels in Korean SLE patients than HC
	McMahon et al., 2011 [44]	Higher levels of leptin in patients with SLE versus controls Higher leptin levels in SLE patients with plaque compared with those without Significantly higher plasma levels of leptin in SLE patients with atherosclerotic plaque Positive correlation between high leptin levels and elevated inflammatory biomarkers such as piHDL, Lp(a) and OxPL/apoB100 High leptin levels correlate with 2.8-fold more atherosclerosis in SLE female patients High leptin levels increase the risk of subclinical atherosclerosis in SLE
	Vadacca et al., 2013 [93]	Higher levels of leptin in SLE patients compared to controls Positive association between leptin levels and vascular stiffness parameters Correlation between leptin levels and disease activity and cumulative damage indexes
	Wang et al., 2017 [94]	Inverse correlation between increased leptin serum levels in SLE patients and the frequency of circulating Tregs Leptin inhibition can expand Tregs
	Diaz-Rizo et al., 2017 [95]	**No correlation between leptin levels in LN patients versus non-LN patients** No correlation between leptin levels and proteinuria severity
	Demir et al., 2018 [96]	Significantly higher leptin levels in SLE patients with MetS versus SLE patients without MetS Correlation between leptin and cIMT levels in premenopausal women with SLE

piHDL = proinflammatory HDL; Lp(a) = Lipoprotein a; oxPL = oxidised phospholipids; MetS = metabolic syndrome; Tregs = regulatory T cells; pSLE = paediatric SLE; HC = healthy control; LN = lupus nephritis.

Studies confirm the presence of a high level of leptin in SLE patients. In SLE, leptin has particular actions on macrophages, favoring the secretion of inflammatory cytokines and increasing oxidative stress. Moreover, it seems to be positively associated with other pro-atherosclerotic and proinflammatory markers. This evidence suggests that leptin may be a promising marker in SLE.

### 3.5. Resistin

Resistin is another adipose tissue-specific secretory marker. It contributes to the inflammatory response by upregulating adhesion molecules, expanding the secretion of proinflammatory cytokines such as IL-1, IL-6, and TNF-α. SLE patients with more severe coronary artery calcification (CAC) showed increased levels of resistin [34,71]. Serum resistin may be an encouraging biomarker for renal involvement in SLE patients, but there are insufficient reports (Table 5).

Studies have shown that resistin is present in an increased titer in lupus patients versus controls. The data support that resistin is involved in the inflammatory process and favors arterial calcification Furthermore, it seems that there is a relationship between resistin and kidney damage in SLE cases.

### 3.6. S100A8 and S100A9

S100A8 and S100A9, two Ca2+ binding proteins, are members of the S100 family. The heterodimeric complex S100 A8/A9 is involved in cytoskeleton rearrangement and arachidonic acid metabolism. It is mainly secreted by phagocytic cells such as neutrophils, monocytes and dendritic cells. S100A8/A9 could be released from activated polymorphonuclear neutrophils (PMNs), as part of NETs [101]. During the inflammatory process, it promotes leukocyte recruitment and induces cytokine secretion [102,103].

Lood et al. evaluated two large cohorts of SLE patients and showed that this proinflammatory and prothrombotic protein complex was found in naïve platelets and has elevated levels in SLE patients. Interestingly, S100A8/A9 is expressed around the thrombus and atherosclerotic plaque and it is synthesized by megakaryocytes, platelet progenitor cells. Wang et al. previously found that this thrombotic effect is mediated through CD36 interaction. This protein complex is associated with CVD, particularly being related to a four-fold increased risk of myocardial infarction [34,103,104].

The serum levels of S100A8/A9 secreted by polymorphonuclear cells are increased in SLE patients, specifically in individuals with positive anti-dsDNA antibodies and glomerulonephritis. Since higher levels of S100A8/A9 have been associated with both inactive and active SLE, serum S100A8/A9 levels may be needed to monitor the disease’s course [101,104,105].

New data consider that S100A8/A9 may also play an interesting role in neuropsychiatric SLE (NPSLE) due to its action on endothelial cells, contributing to vasculopathy and atherosclerosis development at the central nervous system level [106].

Furthermore, S100A8/A9 may be used as a therapeutic biomarker in CVD due to its contribution to atherogenesis, plaque vulnerability, ischemia-associated myocardial inflammation, and heart failure [107] (Table 6). It can generate a proinflammatory response and an increased secretion of IL-6, IL-1β and TNF-α [101]. A promising clinical target may be type-1 interferon (IFN) as it induces the upregulation of platelet-derived S100A8/A9 complex protein [34,103]. S100A8/A9 is becoming a more sensitive biomarker for inflammation activity and for therapeutic response by comparison with traditional inflammation markers, such as C-reactive protein [102,108,109].

This proinflammatory and prothrombotic protein complex S100 A8/A9, especially the one secreted by platelets, is secreted in excess in SLE patients. It is associated with various clinical manifestations of SLE, such as renal or neuropsychiatric damage. It actively participates in the occurrence of CVD by increasing atherogenesis and the vulnerability of the atherosclerotic plaque. Moreover, it favors the secretion of pro-inflammatory cytokines.

### 3.7. NETs and Microparticles (MPs)

Neutrophil extracellular traps (NETs) are chromatin fibers released from dying neutrophils. The death of neutrophils with subsequent NETs production is called “NETosis”. During their activation, neutrophils secrete ROS. The major function of NETs consists of trapping and killing pathogens. NETs exert their antimicrobial mechanism through antimicrobial enzymes such as myeloperoxidase (MPO), neutrophil elastase, but also through other molecules such ascathelicidin, histones and DNA [25,110,111].

The main source of autoantigens in SLE is considered to be the inefficient clearance of necrotic and apoptotic cells. Furthermore, there is a deficient clearing of NETs [2]. Hakkim et al. reported that this impaired degradation of NETs in SLE is apparently due to two mechanisms: the presence of antibodies against endonuclease DNase1—necessary for NETs disassembly and the appearance of anti-NET autoantibodies that prevent NETs from degradation [2,112]. In SLE, the development of NETs is stimulated by ribonucleoprotein-specific antibodies [25,113]. SLE neutrophils are stimulated by type I IFN and die upon exposure to SLE-derived anti-ribonucleoprotein antibodies, releasing NETs [113].

According to the latest research, NETs formation might contribute to atherosclerosis progression [110,114]. NETs can directly cause endothelial cell apoptosis by many mechanisms such as: endothelial cell death mediated by endothelial MMP2 stimulation; activation of platelets, coagulation cascade and thrombosis by releasing serine proteases that deteriorate tissue factor pathway inhibitor and activate factor XII and vascular leakage [25,115,116]. Indirectly, NETs determine low-density granulocytes and plasmacytoid dendritic cells to produce large amounts of IFN, which increases IFN-induced endothelial toxicity, enhances HDL-c oxidation and reduces cholesterol outflow capacity [25,117,118,119]. NETs also determine inflammasome activation, consequently increasing the IL1β and IL-18 production, which finally creates a positive loop of NETs production [2,120]. NETs promote the production and release of type I IFN which further promotes NETosis [2,113]. That is why it has been proposed that NETosis could serve as a therapeutic target in SLE [110].

The production of circulating plasma microparticles (MPs) and NETs are two significant processes that have been described in the context of endothelial dysfunction in SLE (Table 7). MPs are generated from apoptotic cells and have high levels in SLE patients [25,121,122]. Moreover, circulating plasma MP can directly cause endothelial cell death. In addition, MPs and MP-immune complexes can cause alterations in endothelial cell permeability and death by increasing the disruption of endothelial microstructure in SLE [25,123]. MPs are not exclusively linked to endothelial cells; individuals with SLE were found to have high levels of MP produced by platelets, monocytes, granulocytes, and lymphocytes [25,124].

In conclusion, in SLE there is a degradation and deficient clearance of NETs. Thus, through excessive accumulation, atherogenic mechanisms are favored both directly through endothelial cell apoptosis and indirectly through increased IFN secretion, HDL-c oxidation and reduced cholesterol outflow capacity. Moreover, the increase in MPs is associated with the occurrence of CVD due to the increase in vascular permeability and the death of endothelial cells.

### 3.8. Other Potential Biomarkers of Atherosclerosis in SLE

Homocysteine can lead to lipid peroxidation, endothelial dysfunction, and oxidative damage [127,128]. Several studies have associated homocysteine with atherosclerosis in SLE patients [9,45,46,129].

sTWEAK is a potential biomarker for SLE nephritis and CVD in general population. It can increase IFN- expression in peripheral blood mononuclear cells [9,130,131].

Pentraxin-3 (PTX3), a newly identified pro-atherosclerotic marker in SLE, is synthesized by mononuclear phagocytes, myeloid-derived dendritic cells and endothelium cells as a response to local inflammation. It is recognized as a local vascular inflammatory biomarker in SLE [2,132]. It presents high levels in SLE and correlates with disease activity [2,133]. PTX3 is also associated with other indicators of endothelial dysfunction such as soluble VCAM-1 and vWf [2,134]. These findings suggest that PTX3 may be a novel biomarker for early atherosclerosis in SLE.

#### Interferon

The synthesis of lipid mediators, of platelet-activating factors and eicosanoids, antigen presentation, and the synthesis of TNF-α and IL-1 are a small number of pro-atherogenic processes that are upregulated by type II interferon (IFN). IFN -γ is one of the most important cytokine in SLE [2,135,136].

Type 1 IFN, which is mainly secreted by plasmacytoid dendritic cells and low-density granulocytes, plays a crucial role in SLE pathogenesis. Type I IFNs are associated with endothelial dysfunction and with CVD progression in SLE, even in patients without risk factors for vascular diseases [34]. IFN-α has a high toxicity against endothelium [25,137]. IFN-α expression is increased as a result of the activation of Toll-like receptors 7 and 9 [2,138]. Additionally, IFN is involved in abnormal vascular repair because it modulates plaque instability by suppressing the development of smooth muscle cells, endothelial cells and the synthesis of collagen [2]. Activation of IFN pathways is linked to a fast atherosclerosis progression in SLE [69,139]. It remains to be established whether type I IFNs would be able to influence platelet S100A8/A9 levels [103].

TNF-α and IL-1 levels are significantly elevated in SLE patients with CVD. Their role is to promote adhesion molecule expression. TNF-α also contributes to the final stage of atherogenesis through down-regulation of lipoprotein lipase, an enzyme that hydrolyses triglycerides in VLDL (very low density lipoproteins) [2,34,140,141,142].

Interleukin-18 (IL-18) manifests its pro-atherosclerotic and pro-inflammatory effects on endothelial cells. Serum IL-18 levels are increased in SLE and correlate with EPC/CAC (circulating angiogenic cells) dysfunction. In SLE, the increased circulating IFN-α levels determine a chronic dysregulation of inflammasome activity with subsequent increased IL-18 levels that led to an abnormal vascular repair [143,144].

Osteoprotegerin (OPG) belongs to the TNF receptor family. Higher levels of CAC (circulating angiogenic cells) and cIMT have been seen in SLE individuals with higher levels of OPG. Kiani et al. suggested that OPG may be a marker for subclinical atherosclerosis in SLE patients, but it requires additional validation in larger trials [145].

The presence of antiphospholipid antibodies (aPL), which are known to enhance the risk of thrombosis in SLE through a variety of pathways, may also be associated with an accelerated atherosclerosis [2]. Interestingly, Lood et al. reported that aPL antibodies, anti-beta 2 glycoprotein 1 (β2-GP1), anti-cardiolipin and lupus anticoagulant, are correlated with high levels of S100A8/A9 in platelets [103]. aPL interact with endothelial cells and monocytes, creating a pro-inflammatory and pro-coagulant state [2,146,147]. aPL also activate the complement generating C5a; in turn, C5a activates neutrophils and a subsequent extrinsic coagulation cascade [2,148].

In contrast with native LDL, oxLDL can form complexes with IgG β2-GP, also increased in SLE. Moreover, the presence of IgG anti-β2-GP1 enhances OxLDL uptake by macrophages. SLE patients with CVD present increased levels of circulating OxLDL and antibodies against OxLDL. The atherosclerotic process has been positively linked to elevated levels of antibodies against OxLDL epitopes [2,149,150].

The extracellular matrix protein osteopontin (OPN) is considered to be a mediator of systemic inflammation. The role of OPN in SLE and atherosclerosis is linked to type I IFN response regulation [151]. Some studies reported that higher OPN levels correlate with SLE and LN compared with other diseases [152,153,154]. A positive association between OPN and cIMT in SLE patients was also described [155].

Low-density granulocytes (LDGs), a subclass of neutrophils that are frequently found in SLE, are highly susceptible to predispose NETosis. Due to their negative effects on endothelium, due to the synthesis of high levels of pro-inflammatory cytokines such as IFN-α and to the disruption of the EPC development into mature endothelial cells, LDG NETs are believed to contribute to atherosclerosis acceleration in SLE. LDGs also determine an important mitochondrial ROS production. This ROS hyperproduction and the stimulation of lipoproteins peroxidation by a decreased paraoxonase 1 activity, are two additional mechanisms of atherogenesis [2].

SLE patients also have higher levels of specific soluble mediators such as annexin A5, platelet endothelial cell adhesion molecule (PECAM1) and activated leukocyte cell adhesion molecule (ALCAM). These mediators have been linked to endothelial dysfunction [25,156]. Moreover, Valer et al. showed that serum annexin A5 levels were independently correlated with endothelial dysfunction and with an increased carotid intima-media thickness in SLE patients [25,157].

## 4. Conclusions

The balance of endothelial integrity is maintained by the homeostasis between endothelial injury and repair. This is crucial for maintaining a normal endothelial function and consequently, the vascular health. Endothelial dysfunction, which is thought to be the first stage in the pathogenesis of atherosclerosis, develops when endothelial damage exceeds the repair process. It is well known that the atherogenesis process includes both traditional and non-traditional factors, the latter less known, but very important in the mechanism of the disease. Systemic inflammation, also present in SLE, is a main factor in the occurrence of early atherosclerosis, being associated with the presence of a cytokine cascade and multiple cell activations. It is extremely important to find novel biomarkers that can predict the development of atherosclerosis and, why not, to use them as new therapeutic targets.

## Figures and Tables

**Figure 1 ijms-23-12604-f001:**
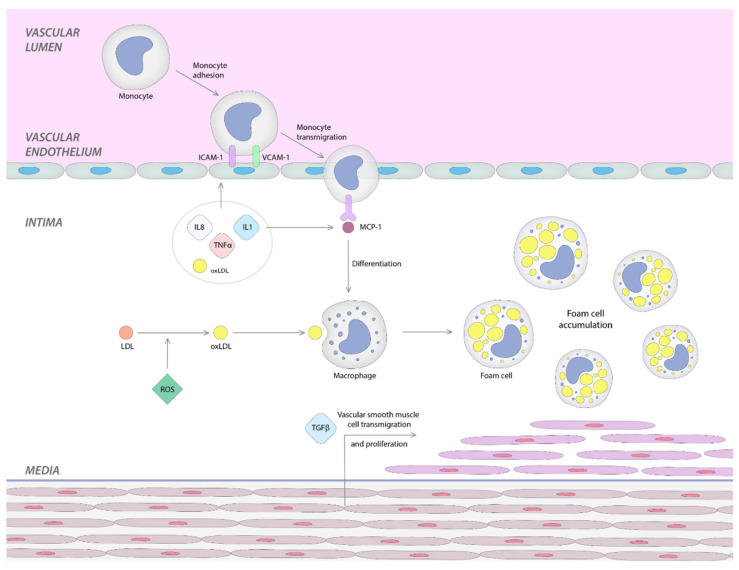
Stages of atherosclerosis mechanism in SLE patients.

**Table 1 ijms-23-12604-t001:** Pro-inflammatory HDL in SLE patients.

Biomarker	Study	Results
**piHDL**	McMahon et al., 2006 [48]	piHDL was found more frequently in SLE patients than in controls and RA patients A higher number of SLE patients had piHLD compared to controls or RA patients (44.7% vs. 4.1% vs. 20.1%) ox-LDL levels correlated with piHDL levels Significantly higher piHDL in SLE patients with CAD than patients without CAD
	McMahon et al., 2009 [49]	piHDL association with higher IMT Strong piHDL association with atherosclerotic plaque on carotid ultrasound
	Skaggs et al., 2010 [55]	Direct role of piHDL on monocyte chemotaxis and secretion of pro-inflammatory molecules MCP-1 and TNFα Direct alteration of a small number of transcripts in circulating monocytes HDL inhibition restored monocyte chemotaxis

RA = rheumatoid arthritis; HDL = high-density lipoproteins; oxLDL = oxidized low-density lipoprotein; CAD = coronary artery disease; MCP-1 = monocyte chemoattractant protein-1; TNF = tumor necrosis factor; IMT = intima-media thickness.

**Table 2 ijms-23-12604-t002:** Endothelial progenitor cells and the relation with SLE manifestations.

Biomarker	Study	Results
**EPCs**	Westerweel et al., 2007 [67]	Lower levels of EPCs in SLE patients in clinical remission
	Moonen et al., 2007 [68]	Reduced number of CD34–CD133 double-positive CPC cells in SLE patients
	Denny et al., 2007 [69]	Significantly lower circulating EPC levels in SLE patients Correlation between decreased EPC numbers and SLEDAI; also in patients with no clinical or serologic disease activity (SLEDAI = 0) Reduced EPC may significantly affect vascular repair
	Lee et al., 2007 [70]	SLE patients had notably decreased levels of EPC colony-forming units compared to controls Elevated levels of IFN-I showed an important depletion of EPCs in SLE patients CRP independently associated with the decrease of EPCs
	Baker et al., 2012 [71]	Significantly lower number of EPCs in SLE patients with no evidence of CAC
	Kim et al., 2013 [72]	Reduced number of EPCs in SLE patients High levels of OPG correlate with low levels of EPCs OPG may induce EPCs apoptosis
	Castejon et al., 2014 [66]	Association between lower levels of EPCs and high arterial stiffness and other CV risk factors, such as tobacco use or metabolic syndrome
	Huang et al., 2021 [73]	Significantly reduced CD34+ CD133+ CD309+ CAC level in SLE patients versus HC Notable correlation between HCQ use and a higher level of CAC

EPC = Endothelial progenitor cells; SLEDAI = Systemic Lupus Erythematosus Disease Activity Index; CAC = coronary artery calcification; CRP = C-reactive protein; IFN = interferon; CPC = circulating progenitor cells; OPG = osteoprotegerin; CV = cardiovascular; HCQ = hydroxychloroquine.

**Table 3 ijms-23-12604-t003:** Endocan in SLE cases.

Biomarker	Study	Results
**Endocan**	Icli et al., 2016 [76]	Positive correlation between endocan levels and cIMT
	Tokarska et al., 2020 [77]	Significantly association between high endocan serum concentration in SLE patients versus HC Notable correlation between serum levels of endocan and IL-1β, suggesting that a high endocan level may be a biomarker of endothelial dysfunction in SLE

cIMT = carotid intima-media thickness; HC = healthy controls; IL = interleukin.

**Table 5 ijms-23-12604-t005:** Resistin in SLE patients.

Biomarker	Study	Results
**Resistin**	Baker et al., 2011 [97]	Higher resistin levels in SLE patients with CAC compared to SLE without CAC Positive correlation between high resistin levels and markers of inflammation and renal dysfunction in SLE patients No correlation with insulin resistance
	Hutcheson et al., 2015 [98]	Increased resistin levels in sera and urine of LN patients Correlation between serum resistin and renal dysfunction in LN
	Chougule et al., 2018 [84]	Statistically significant elevation of resistin levels in SLE patients versus HC High resistin levels in SLE patients with renal manifestations
	Gamez-Nava et al., 2020 [99]	Higher resistin levels in SLE patients with renal involvement versus patients without renal SLE Association between resistin and proteinuria Resistin correlations with increased creatinine levels
	Shaaban et al., 2021 [100]	Significantly association between serum resistin and hs-CRP, HDL and anti-nuclear antibody No significantly elevated levels of resistin in SLE compared to controls

CAC = coronary artery calcification; LN = lupus nephritis; HC = healthy controls; hs-CRP = high-sensitivity C-reactive protein; HDL = high-density lipoprotein.

**Table 6 ijms-23-12604-t006:** Protein complex S100 A8/A9 in SLE patients.

Biomarker	Study	Results
**Protein complex S100 A8/A9**	Soyfoo et al., 2009 [108]	Significantly elevated S100A8/A9 serum levels in SLE patients compared to pSS patients and HC Positive correlation between S100A8/A9 and SLEDAI S100A8/A9 may also be considered a marker of infection, as SLE patients with infections presented elevated serum levels of S100A8/A9
	Tydén et al., 2013 [101]	Increased serum levels of S100A8/A9 in SLE patients with inactive disease (SLEDAI = 0) versus controls Importantly increased serum levels of S100A8/A9 in SLE patients with a history of acute myocardial infarction or cerebrovascular complications compared to SLE patients without CVD or venous thromboembolism Association between serum levels of S100A8/A9 and disease severity and organ damage, independently of CVD Correlation between anti-dsDNA and elevated S100A8/A9 concentrations
	Lood et al., 2016 [103]	Increased platelet S100A8/A9 expression in SLE when compared with controls Specifically, high S100A8/A9 protein levels in platelets from SLE patients Association between increased plasma levels of S100A8/A9 with platelet activation in SLE patients Strong association between high levels of platelet S100A8/A9 and CVD such as myocardial infarction or APS
	Tydén et al., 2016 [105]	High serum levels of S100A8/A9 in SLE patients with high disease activity versus patients with low disease activity Increased levels of S100A8/A9 in patients with anti-dsDNA antibodies and GN before treatment

pSS = primary Sjögren’s syndrome; HC = healthy controls; CVD = cardiovascular disease; SLEDAI = Systemic Lupus Erythematosus Disease Activity Index; APS = antiphosolipidic syndrome; anti-dsDNA = anti-double stranded DNA; GN = glomerulonephritis.

**Table 7 ijms-23-12604-t007:** Microparticles in SLE cases.

Biomarker	Study	Results
**MPs**	Mobarrez et al., 2016 [121]	MP are 2-10 times more abundant in SLE blood compared to controls
	Dieker et al., 2016 [125]	Plasma MP from SLE patients exerts significant effects on MDCs and PDCs by stimulating the production of proinflammatory cytokines (IL-6, TNF, and ITN-α) MP enhances NETosis by blood-derived neutrophils interaction
	Atehortúa et al., 2019 [123]	MPs and MPs-ICs from patients with SLE and RA induce structural modifications on macrovascular and microvascular endothelial cells by increasing adhesion molecules expression and chemokine production
	Carmona-Pérez et al., 2021 [126]	Significantly elevated CD14+ MPs in patients with SLE than HCs Positive correlation between MPs and SLEDAI score Increased plasma MPs concentration in patients with active SLE

MPs = microparticles; IC = immune complexes; RA = rheumatoid arthritis; IL-6 = interleukin-6; TNF = tumor necrosis factor; ITN-α = interferon-α; PDCs = blood-derived plasmacytoid dendritic cells; MDCs = myeloid dendritic cells; SLEDAI = SLE Disease Activity Index.

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
