# Peer review of "Non-Traditional Pro-Inflammatory and Pro-Atherosclerotic Risk Factors Related to Systemic Lupus Erythematosus"

_ijms, 2022, doi:10.3390/ijms232012604_

Round 1

Reviewer 1 Report

The work is sound, well written and add significant contribution to the field. However, I would suggest to summarize each subsection with short paragraph of 4-5 sentences, that could be useful for the readers to follow.

Author Response

Dear Reviewer,

Thank you very much for the attention given to this work and for the requested changes that surely improved the quality of the article. Following the instructions received, I made the following changes, as follows:

  • we added a small conclusion at the end of each subchapter

We hope that these changes satisfy your requirements and we look forward to other suggestions if you consider them appropriate.

Sincerely,

The authors

Reviewer 2 Report

This review is instresting and well written. Here, the author stated the progress for non-traditional pro-inflammatory and pro-atherosclerotic risk factors related to SLE, however, the followed questions need to be cosdidered:

1.In addition to tables, figures to show the mechanisms are still needed.

2.In addition to SLE patients, what about these biomaikers in the animal models for lupus, especially for very few studies in SLE patients, e.g. Endocan.

Author Response

Dear Reviewer,

Thank you very much for the attention given to this work and for the requested changes that surely improved the quality of the article. Following the instructions received, I made the following changes, as follows:

  • a figure would help readers to better understand the mechanism involved in the inflammation-atherosclerosis link. We found this a very good idea, so we added figure 1.
  • we have added some data regarding endocan and endothelial progenitor cells on animal models

We hope that these changes satisfy your requirements and we look forward to other suggestions if you consider them appropriate.

Sincerely,

The authors

Reviewer 3 Report

In this manuscript, the authors have reviewed a large amount of literature and listed risk factors associated with the higher incidence of atherosclerosis in patients with SLE. Evidence linking several of these risk factors (pro-inflammatory HDL, endothelial progenitor cells, endocan, leptin whose association with atherosclerosis is controversial, resistin, S100A8/A9 and microparticles) are conveniently presented as Tables with references and with the main findings and conclusions. Further risk factors are mentioned in the text. When known, the molecular link between these risk factors and atherosclerosis is mentioned, but for others, it is more like a mere list.

I would suggest to add a short paragraph describing the strategy applied to mine and select the literature cited in this review.

LES disease is mentioned 5 times (lines 117, 175, 184, 198 and 338), but this abbreviation is not defined. I suspect it is an alternative way of writing SLE. If true, then please change to SLE.

There are a few typos and sentences to clarify. I saw them, I mention them: Line 34: "the risk is twice as higher" --> "the risk is twice as high". Line 98: "Thus, it has been created PREDICTS model." --> "Thus, a model was created that is called PREDICTS". Line 119: "Moreover, piHDL function might show proteomic and lipidomic abnormalities". I cannot understand how a function can have proteomic abnormalities. Wrtie "Moreover, piHDL have proteomic and lipidomic abnormalities", or rewrite the sentence. Table 4, reference McMahon 2011: "high leptin levels enhance atherosclerosis in SLE female patients by 2.8-fold". According to text of lines 163-167, how leptin leads to this enhancement is not really known. If ture, write "high leptin levels correlate with 2.8-fold more atherosclerosis in SLE female patients". Table 4, last reference Demir et al: "SLE patients with MetS versus SLE MetS". Do authors mean "SLE patients with MetS versus SLE patients without MetS"?. Line 192: "members of S100 family" -> "members of the S100 family". Line 230-231: "NETS exert their antimicrobial mechanism through anti-microbial enzymes such as MPO, neutrophil elastase, cathelicidin, histones and DNA". The fisrt two are indeed enzymes, but the later three are not. Please rephrase. Line 255: "MPs are generated from apoptotic cells and express high levels in SLE patients". Express high levels of what? Or do author mean that MPs are produced at high levels in SLE patients? Please clarify. Line 296-297: "TNF also contributes to the final stage of atherogenesis due to the inhibition of LPL, an enzyme that prevents triglyceride and very LDL metabolization". Unclear. "An enzyme" probably refers to LPL. But LPL hydrolyses triglycerides in VLDL, and does not prevent it. What is very LDL? VLDL? (if yes, use VLDL). Also the formulation suggest that TNF is a direct inhibitor of LPL, which is surely not the case. What about "TNF also contributes to the final stage of atherogenesis through down-regulation of lipoprotein lipase, an enzyme that hydrolyses triglycerides in VLDL (very low density lipoproteins)". Line 299: CAC means circulating angiogenic cells, while CAC means coronary arthery calcification in line 149, 185, 189. It is not very convenient to have the same abbreviation for 2 different things. Fortunately, this causes no ambiguity becuase the full name is always given, except at line 303 wher I suggest to add "circulating angiogenic cells" again to avoid confusion. Lines 311 and 312. The abbreviation used for anti-phospholipids is "aPL". Do not capitalize the "a" at the beginning of a sentence (I wondered what was the meaning of APL...). Line 316: "IgG beta2-GP1". Do authors mean "IgG anti-beta2 GP1"?". Lines 323-324: "It was also described positive association between ..." --> "A positive association between ... was also described"

Author Response

Dear Reviewer,

Thank you very much for the attention given to this work and for the requested changes that surely improved the quality of the article. Following the instructions received, I made the following changes, as follows:

- we added a paragraph regarding the work strategy and data collection

- we changed LES to SLE throughout the work

- we have changed all the typos according to the received suggestions

We hope that these changes satisfy your requirements and we look forward to other suggestions if you consider them appropriate.

Sincerely,

The authors